# Online learning of long-range dependencies

**Nicolas Zucchet**\*, **Robert Meier**\*, **Simon Schug**\*
**Asier Mujika**, **João Sacramento**

Department of Computer Science, ETH Zürich
`{nzucchet,romeier,sschug,asierm,rjoao}@ethz.ch`

## Abstract

Online learning holds the promise of enabling efficient long-term credit assignment in recurrent neural networks. However, current algorithms fall short of offline backpropagation by either not being scalable or failing to learn long-range dependencies. Here we present a high-performance online learning algorithm that merely doubles the memory and computational requirements of a single inference pass. We achieve this by leveraging independent recurrent modules in multi-layer networks, an architectural motif that has recently been shown to be particularly powerful. Experiments on synthetic memory problems and on the challenging long-range arena benchmark suite reveal that our algorithm performs competitively, establishing a new standard for what can be achieved through online learning. This ability to learn long-range dependencies offers a new perspective on learning in the brain and opens a promising avenue in neuromorphic computing.

## 1 Introduction

How can the connections between neurons in a neural network be adjusted to improve behavior? This question, known as the credit assignment problem, is central in both neuroscience [1] and machine learning [2], owing to its fundamental importance for elucidating learning mechanisms in the brain and constructing intelligent artificial systems. However, the complex and nonlinear nature of neural network processing makes the precise allocation of credit an intricate task.

Deep learning provides a compelling solution to the credit assignment problem via gradient descent, which refines network parameters along the locally most promising direction. For networks processing temporal sequences, gradient computation is made possible by backpropagation-through-time [BPTT; 2–4]. BPTT stores and revisits neural activity in reverse-time order to understand how infinitesimal changes to neural activity, and thus to network parameters, would have impacted the objective function. One important drawback of this algorithm is its requirement to store the entire activity trajectory in memory, which constrains the sequence length for exact gradient computation, impairing the learning of long-term interactions. This constraint becomes a critical bottleneck when working within memory-limited systems, such as neuromorphic hardware [5] and presumably the brain [6].

Alternatives to BPTT for gradient computation do exist. One such approach, forward-mode differentiation [7, 8], involves computing gradients online as the input sequence is processed, by keeping track of the sensitivity of neural activity with respect to each of the network parameters. This marks a qualitative departure from BPTT, as it prepares for all potential future trajectories simultaneously; by contrast, BPTT focuses on improving the activity of a past trajectory. Importantly, the memory footprint of this approach does not depend on sequence length. Still, it remains intractable for real-world applications and likely infeasible in the brain due to its cubic memory scaling and quartic computational complexity in the number of neurons. Recent research focused on approximation

---

\*Equal contribution.

37th Conference on Neural Information Processing Systems (NeurIPS 2023).

strategies to make online gradient estimation more tractable for general-purpose recurrent networks [9–18]. Our work takes a fundamentally different approach: instead of tailoring the learning algorithm to the neural network architecture, we fix the learning algorithm and seek an architecture that makes it tractable.

We build upon recent advances in linear state space models, a class of recurrent neural networks (RNNs) employing linear recurrent blocks [19–23]. These blocks are stacked and interconnected through nonlinear networks. The key insights from this line of research are that linear recurrent connections simplify temporal credit assignment and enable parallel temporal processing, while nonlinearities between recurrent blocks ensure that network expressiveness remains comparable to that of densely-connected nonlinear RNNs. Much, if not all, of the state-of-the-art performance of those models on long-range temporal tasks [24] can be maintained by transitioning from real-valued to complex-valued neural activities [21–23], and restricting the recurrent connectivity matrix to be diagonal. Recurrent neurons within a given layer are now independent of each other. This greatly improves the tractability of online gradient estimation, as the recurrent parameters of a given neuron do not impact other neurons. We leverage this property to achieve exact online gradient computation within a single layer with as little as twice the memory and compute requirements needed for inference. Further, we demonstrate how this leads to improved gradient estimation compared to existing online learning algorithms when recurrent layers are stacked.

This paper is organized as follows. We start by briefly reviewing existing gradient-based online learning methods in Section 2.1. Next, we introduce the concept of independent recurrent modules, showing how some of the recent high-performance models mentioned above fit in this framework in Section 2.2. Deriving our learning rule requires complex differentiation; we give a concise overview of those tools in Section 2.3. In Section 3, we detail our online learning algorithm that combines exact differentiation within a layer of recurrent independent modules with spatial backpropagation across layers. Finally, in Section 4, we analyze our algorithm and relevant baselines on a synthetic copy task and show that it can learn sequential tasks with sequence lengths up to over $4000$ steps.

## 2 Background

### 2.1 Online gradient-based RNN learning

We study gradient-based learning of recurrent neural networks, which process input data $x_1, \ldots, x_T$ sequentially while maintaining an internal (hidden) state $h_t$. The objective of learning is to minimize a cumulative loss $L(\theta) = \sum_{t=1}^{T} L_t(\theta)$ which measures performance on a task at hand as a function of network parameters $\theta$. The standard algorithm for computing the gradient $\nabla L(\theta)$ is the offline backpropagation-through-time method, which requires storing the entire input $x_{1:T}$, loss $L_{1:T}$ and internal activity $h_{1:T}$ sequences, and then revisiting them proceeding backwards in time. Here, we focus on online algorithms which carry the information needed to compute or estimate $\nabla L_t(\theta)$ forward in time. This enables simultaneous processing of inputs and learning for RNNs, without storing past data and network states. In principle, online algorithms can learn arbitrarily long temporal dependencies as well as seamlessly handle sequences of arbitrary length $T$.

The classical alternative to BPTT for forward-in-time gradient computation is known as real-time recurrent learning [RTRL; 7] in the context of RNNs[2], a method which has its roots in control theory [26]. While RTRL enables online gradient-based learning, it requires storing $d_\theta h_t$ in memory and updating it as the RNN processes its inputs. The size of this auxiliary variable is $\mathcal{O}(n^3)$, where $n = |h|$ is the number of hidden units in the RNN. For comparison, the memory requirements of BPTT are $\mathcal{O}(nT)$. In practice, this precludes the usage of RTRL for all but the smallest of models.

There has been much effort in developing memory-efficient alternatives to RTRL. We now briefly discuss these prior efforts while referring to a recent review by Marschall et al. [14] for a more detailed treatment. We divide prior work into three broad categories. One class of algorithms relies on neglecting terms in $d_\theta h_t$ to reduce its size, thereby creating a biased gradient estimator. Typically this requires introducing crude approximations, which allow going from $\mathcal{O}(n^3)$ to a tractable $\mathcal{O}(n^2)$ size. Despite such approximations, in many cases, performance still holds in non-trivial tasks [13, 15, 16]. At the end of this spectrum sits instantaneous (spatial) backpropagation, which neglects all temporal

---

[2]More generally, BPTT can be seen as a special case of reverse-mode automatic differentiation, and RTRL of forward-mode automatic differentiation [25], applied to the problem of RNN learning.

dependencies in the hidden state when approximating the gradient. A second class of algorithms relies on stochastic estimation; this allows retaining unbiased gradient estimates, at the expense of introducing variance [9–11, 17, 18]. Finally, a third class of methods introduces gradient models [critics; 27–29] to produce gradient estimates online. The critics themselves are then either trained separately offline, making such methods hybrid on/offline; or fully online, using temporal difference techniques [30]. We note that despite their widespread use in reinforcement learning, it is not yet well understood whether temporal difference methods can reliably and efficiently improve the performance of a gradient critic on real-world RNN learning problems.

As noted by Irie et al. [31], the RTRL literature mostly focuses on single-layer recurrent networks and remains scarce for deeper networks. Recently, Javed et al. [32] developed a greedy learning algorithm where a growing network is progressively frozen and trained one layer at a time. Existing approximations such as [13, 15] do not prescribe how to learn the parameters of remote layers, and the multi-layer case is not considered in the respective papers. Introducing a powerful RTRL algorithm that scales to networks of multiple layers is the main algorithmic contribution of this paper. This property is of great empirical relevance given the power of depth to learn the temporal structure [e.g. 19, 23].

## 2.2 Linear recurrent units and independent recurrent modules

Instead of developing approximate, general-purpose forward-mode differentiation methods, our goal shifts towards seeking an expressive architecture allowing exact, tractable online gradient calculation. We propose that networks with linear recurrent units [LRU; 23] and, more generally networks with independent recurrent modules, are particularly well-suited for this purpose.

A linear recurrent unit, depicted in Figure 1, is defined as

$$h_{t+1} = \lambda \odot h_t + Bx_{t+1}, \quad y_t = \text{Re}[Ch_t] + Dx_t, \tag{1}$$

with $\odot$ the element-wise product. Here, $x_t \in \mathbb{R}^H$ represents the input received by the LRU at time $t$, $h_t \in \mathbb{C}^N$ denotes its internal state, and $y_t \in \mathbb{R}^H$ its output. The parameters of the unit include $\lambda \in \mathbb{C}^N$, $B \in \mathbb{C}^{N \times H}$, $C \in \mathbb{C}^{H \times N}$ and $D \in \mathbb{R}^{H \times H}$. The version of the LRU we use in our experiments includes an element-wise normalization factor for the input $Bx$ and uses an exponential parametrization of $\lambda$ for network stability. We omit these details in the main text for conciseness; see Appendix A.1 for more details.

LRUs differ from traditional recurrent layers in deep learning: they have linear neural dynamics, a complex-valued state $h_t$, and a diagonal connectivity pattern. Orvieto et al. [23] found that the absence of temporal linearity in networks that stack those units through nonlinear connections (see Fig. 1) does not alter expressivity and eases gradient-based learning, a notoriously difficult process for nonlinear RNNs [33, 34]. In addition, the diagonal structure of the recurrence matrix provides several benefits over fully connected ones. First, it affords an easy way to control the eigenvalues of the Jacobian of the system and thus ensure that neural dynamics remain stable. Due to the linearity, it also enables processing the input sequence in parallel [35], significantly accelerating the training of such models on modern computers [22]. Importantly, despite its diagonal parametrization, the LRU remains functionally equivalent to a linear recurrent layer with dense recurrence matrix $A$, as $A$ can be approximated as accurately as needed by a complex-diagonalizable matrix.

Each complex neuron in an LRU is an *independent recurrent module*, meaning its current state does not impact the dynamics of other modules. This property greatly simplifies online credit assignment (see Section 3). We focus on this specific architecture due to its simplicity and great empirical performance, but our theoretical insights also apply to networks of independent recurrent modules with low-dimensional state vectors per module.

## 2.3 A primer on complex differentiation

The use of complex-valued networks, such as the LRU, and hence complex differentiation remains relatively scarce. In the following, we provide a concise review of the tools of complex differentiation integral to the derivation of our online learning rule. We use $f$ and $g$ to denote complex-valued functions that take the complex variable $z$ as input.

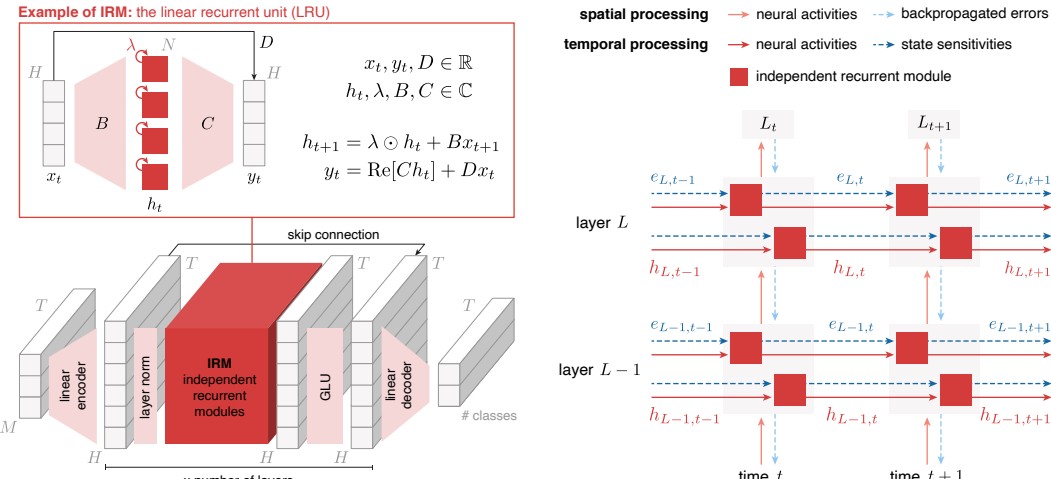

Figure 1: (Left) Overview of the class of neural networks we consider in this paper. We stack layers of independent recurrent modules (IRMs), augmented with layer norm [36] and gated linear units [GLU; 37]. Light red indicates instantaneous spatial processing, dark red temporal processing. When we examine networks with fully connected recurrent layers, only the dark red block is modified. The linear recurrent unit is the instantiation of a layer of IRMs we use in our experiments. (Right) Overview of our learning rule. As an input sequence is processed, hidden states $h_t$ and their sensitivities $e_t$ to the parameters are updated. Learning ensues by combining the sensitivities $e_t$ with spatially backpropagated error signals. No information flows in reverse time; our rule is fully online.

The Wirtinger derivatives of $f$ are defined through

$$\frac{\mathrm{d}f}{\mathrm{d}z} := \frac{1}{2}\left(\frac{\mathrm{d}f}{\mathrm{d}\mathrm{Re}[z]} - i\frac{\mathrm{d}f}{\mathrm{d}\mathrm{Im}[z]}\right), \ \frac{\mathrm{d}f}{\mathrm{d}\bar{z}} := \frac{1}{2}\left(\frac{\mathrm{d}f}{\mathrm{d}\mathrm{Re}[z]} + i\frac{\mathrm{d}f}{\mathrm{d}\mathrm{Im}[z]}\right). \quad (2)$$

Using them for complex differentiation allows using similar calculus rules as for real functions. Note that we use the row convention for derivatives, that is $\mathrm{d}_z f$ is a row vector of size $|z|$. The following formula holds in general $\overline{\mathrm{d}_z f} = \mathrm{d}_{\bar{z}}\bar{f}$.

The complex derivative of a complex function is similar to a $2 \times 2$ real-valued matrix as both $\mathrm{d}_z f \in \mathbb{C}$ and $\mathrm{d}_{\bar{z}} f \in \mathbb{C}$ are necessary to characterize it. Yet, there exists a subclass of functions, called holomorphic functions for which it can be reduced to a 2 dimensional real-valued vector, leading to a more compact representation of derivatives. A continuous function $f$ is *holomorphic* if it satisfies the Cauchy-Riemann equations

$$\frac{\mathrm{d}\mathrm{Re}[f]}{\mathrm{d}\mathrm{Re}[z]} = \frac{\mathrm{d}\mathrm{Im}[f]}{\mathrm{d}\mathrm{Im}[z]} \ \text{ and } \ \frac{\mathrm{d}\mathrm{Re}[f]}{\mathrm{d}\mathrm{Im}[z]} = -\frac{\mathrm{d}\mathrm{Im}[f]}{\mathrm{d}\mathrm{Re}[z]}, \ \text{ i.e. } \frac{\mathrm{d}f}{\mathrm{d}\bar{z}} = 0. \quad (3)$$

Any affine function, as well as the composition of two holomorphic functions, are themselves holomorphic.

The chain rule of complex differentiation, crucial for automatic differentiation, is

$$\frac{\mathrm{d}(f \circ g)}{\mathrm{d}z} = \frac{\mathrm{d}f}{\mathrm{d}g}\frac{\mathrm{d}g}{\mathrm{d}z} + \frac{\mathrm{d}f}{\mathrm{d}\bar{g}}\frac{\mathrm{d}\bar{g}}{\mathrm{d}z}. \quad (4)$$

When either $f$ or $g$ is holomorphic, the second term vanishes as $d_{\bar{g}}f = 0$ or $\mathrm{d}_z\bar{g} = \overline{\mathrm{d}_{\bar{z}}g} = 0$. When $f$ is a real-valued function, it can be optimized through gradient descent by iteratively updating its input variable z through $\Delta z \propto -\mathrm{d}_{\mathrm{Re}[z]}f^\top - i\mathrm{d}_{\mathrm{Im}[z]}f^\top = -2\mathrm{d}_{\bar{z}}f^\top$.

## 3 Online learning of networks of independent recurrent modules

Based on the foundations laid down in the preceding section we now derive our online gradient-based learning algorithm for multi-layer networks of independent recurrent modules. We first focus on a single layer, demonstrating that exact forward-mode differentiation is tractable. This insight then guides the derivation of our rule for multi-layer networks.

## 3.1 Single-layer networks

We focus on parameters $\theta$ that influence the hidden states; computing the gradient of any other parameter does not require temporal credit assignment. For the LRU, we have $\theta = \{\lambda, B\}$. Recall that $L(\theta) = \sum_{t=1}^{T} L_t(y_t(\theta))$ denotes the loss function that measures how good the outputs $y_{1:T}$ of a network parametrized by $\theta$ are. Its derivative $\mathrm{d}_\theta L(\theta)$ can be calculated using forward-mode (complex) differentiation:

$$\frac{\mathrm{d}L}{\mathrm{d}\theta} = \sum_{t=1}^{T} \frac{\partial L}{\partial h_t} \frac{\mathrm{d}h_t}{\mathrm{d}\theta} + \frac{\partial L}{\mathrm{d}\overline{h_t}} \frac{\mathrm{d}\overline{h_t}}{\mathrm{d}\theta} = \sum_{t=1}^{T} \frac{\partial L_t}{\partial h_t} \frac{\mathrm{d}h_t}{\mathrm{d}\theta}. \tag{5}$$

As mentioned in Section 2.3, the last equality holds as $h_{t+1}$ is a holomorphic function of $h_t$ and of $\theta$, hence $h_t$ is a holomorphic function of $\theta$ by recursive composition, and $L$ only directly depends on $h_t$ through $L_t$. The term $\delta_t := \mathrm{d}_{h_t} L^\top$ that is here equal to $\mathrm{d}_{h_t} L_t^\top$ can easily be computed by spatial backpropagation, as the output $y_t$ at time $t$ only depends on the current hidden state $h_t$. We are left with computing the sensitivities $\mathrm{d}_\theta h_t$ of the states to the parameters.

Independent recurrent modules do not influence each other. The parameters $\theta_i$ that directly influence the state $h_{t,i}$ of module $i$ never impact the state $h_{t',j}$ of another module. As a consequence, the number of non-zero entries of the sensitivity $\mathrm{d}_\theta h_t$ grows linearly with the size of $\theta$ whenever the number of recurrent neurons within each module is fixed. Applying this to the LRU, $\mathrm{d}_\theta h_t$ is entirely characterized by $e_t^\lambda := (\mathrm{d}_{\lambda_i} h_{t,i})_i$ and $e_t^B := (\mathrm{d}_{B_{ji}} h_{t,j})_{ji}$. Differentiating Equation 1 using the product rule gives the corresponding updates:

$$e_{t+1}^\lambda = \lambda \odot e_t^\lambda + h_t, \;\; e_{t+1}^B = \mathrm{diag}(\lambda) e_t^B + 1 x_{t+1}^\top, \tag{6}$$

with 1 a vector of size $|h|$ filled with ones. More detail on the derivation of Equation 6 and on how to efficiently simulate this update are given in Appendix A.2. Keeping track of those quantities only requires considering an additional hidden state of size $|\theta|$.[3] Finally, the $\lambda$ and $B$ updates can be obtained by following the gradient, as calculated in Equation 5:

$$\Delta\lambda \propto \sum_{t=1}^{T} \delta_t \odot e_t^\lambda, \;\; \Delta B \propto \sum_{t=1}^{T} \mathrm{diag}(\delta_t) e_t^B. \tag{7}$$

Interestingly, all the $e$-updates are local to the neuron or synapse in question, and no approximations were required to accomplish this. This feature makes the algorithm particularly promising for neuroscience and neuromorphic engineering, where localized computation is highly desirable. The parameter update for $\lambda$ and $B$ is also fully local, as it combines a parameter-specific sensitivity, sometimes considered as an eligibility trace [13], and a postsynaptic error term.

The idea that element-wise recurrence simplifies RTRL precedes our work. It can be found in early work by Mozer [38] and Gori et al. [39], and has been revisited recently [32, 31]. In this paper, we extend this insight to complex numbers and thus do not lose expressivity, unlike previous work. We also note that some approximations to RTRL such as e-prop [13] or SnAp-1 [15] end up being exact when applied to networks with independent recurrent modules.

## 3.2 Multi-layer networks

The derivation in the last section presumes that the loss $L$ only directly depends on $h_t$ through $L_t$. This assumption no longer holds when layers are stacked, which is crucial to the expressiveness of the model. In the following, we explain how we can extend our rule to the multilayer case. Let us consider layer $l$ of the network where we aim to compute the gradient of the loss $L$ with respect to its parameters $\theta^l$. The sensitivity $\mathrm{d}_{\theta^l} h_t^l$ can be computed as before, assuming independent recurrent modules, as $\theta^l$ does not influence the behavior of the inputs it receives from previous layers. Hence, we are left with computing $\delta_t^l = \mathrm{d}_{h_t^l} L^\top$. The simplification $\delta_t^l = \mathrm{d}_{h_t^l} L_t^\top$ we made in the previous section still holds for the last layer but is violated for the other layers. This is because $L$, taken as function of the hidden states $h^l$ of layer $l$, now has an internal memory through the subsequent recurrent layers. The hidden state $h_t^l$ at time $t$ will thus directly affect all future losses $L_{t'}$ for $t' \geq t$.

---

[3]If $h_t$ is not a holomorphic function of $\theta$, one would need to keep $2|\theta|$ instead of $|\theta|$ states.

As a consequence, one has to resort to backpropagation-through-time to compute $\mathrm{d}_{h_t^l} L^\top$ exactly, which breaks causality and rules out the possibility of learning online. To circumvent this issue, we approximate the error signal each layer receives by $\delta_t^l \approx \mathrm{d}_{h_t^l} L_t^\top$ so that it can be computed instantaneously with spatial backpropagation. We emphasize that the only source of approximation of this algorithm is the one above. Given that there is no approximation for the last layer, we will always compute the exact gradient for that layer.

We summarize our learning rule in Figure 1. It prescribes augmenting the hidden state of each recurrent layer $l$ with the sensitivity $e^l$. For each input/output sample $(x_t, y_t^{\mathrm{target}})$, we first update the full entire hidden state $(h_t, e_t)$ using the previous one $(h_{t-1}, e_{t-1})$ and current input $x_t$. We then spatially backpropagate the error signal obtained at the last layer by comparing the prediction of the network to its desired value $y_t^{\mathrm{target}}$. Finally, we combine $e_t^l$ and $\delta_t^l$ available at each layer using Equation 7 to compute the current update. So far, we have only described how to update parameters that directly influence the hidden neurons of the recurrent layer. We update the rest of the parameters with the gradient estimate obtained with spatial backpropagation. Importantly, the size of the extended hidden state is upper bounded by the number of neurons plus the number of parameters and is in practice much smaller.

Next, we aim to understand the factors that may degrade the quality of the gradient estimate as a result of the approximation $\delta_t^l \approx \mathrm{d}_{h_t} L_t^\top$ which introduces bias in the gradient estimate. In the LRU, neural activities, and thus error signals, are processed temporally through dynamics similar to the one of Equation 1. When the norm of each $\lambda_i$ approaches one, neural activity preserves past information, and correspondingly, error signals backpropagated over time contain more information about the future. This suggests that our approximation becomes less accurate as the distribution of $|\lambda_i|$ narrows around 1, since it discards future error information. Moreover, $\delta$ worsens as it is backpropagated through more layers. At each layer, backpropagation mixes errors from the next layer and the future state of the layer. Since we neglect future information, only part of the error signal is backpropagated, resulting in a less accurate approximation. We delve deeper into these two approximation sources in a memory task in Section 4.1.

# 4    Experiments

In the following, we analyze the properties of our proposed online learning method empirically and explore how independent recurrent modules aid learning of long-range dependencies. To this end, we first conduct an extensive analysis on the copy task [40], a well-established test bed to study temporal credit assignment in recurrent neural networks. Comparisons to truncated online versions of BPTT and to an extension of SnAp-1 to deep recurrent networks, reveal that independent recurrent modules are generally beneficial for learning long-range dependencies, as they excel in both the online and the offline setting. Finally, we evaluate our method on three tasks of the Long Range Arena [24]: a sequential version of CIFAR [41], LISTOPS [42] and IMDB [43], scaling online learning to sequence lengths of over $4000$ time steps and to deep recurrent neural networks. For additional experimental details and hyperparameter configurations, we refer to Appendix B.

## 4.1    Understanding the approximations behind online learning in networks of LRUs

In this first series of experiments, we investigate the approximation introduced by our online algorithm in detail. We recall that our learning rule only approximates the full gradient when the network has two recurrent layers or more, as it ignores the temporal component of backpropagated error signals. Therefore, we expect that the learning signal becomes less accurate as we increase network depth and shift the eigenvalue distribution towards a norm of $1$. To explore the effects of our approximation in a controlled setting, we consider a copy task [40, 10, 15, 18] in which the network observes a length-20 sequence of 7-bit patterns that it must recall when presented with an output token. Intuitively, temporal credit assignment is critical in this task to, among other things, convert forgetting neurons ($|\lambda| \ll 1$) into perfect memory units ($|\lambda| \approx 1$) that can solve the task. To ensure that these perfect memory units are scarce at the beginning of training, necessitating learning to create some, we initialize $\lambda$ uniformly at random in the complex unit disk and set the number of recurrent units per layer to $N = 64$. The default architecture used in this section has four layers.

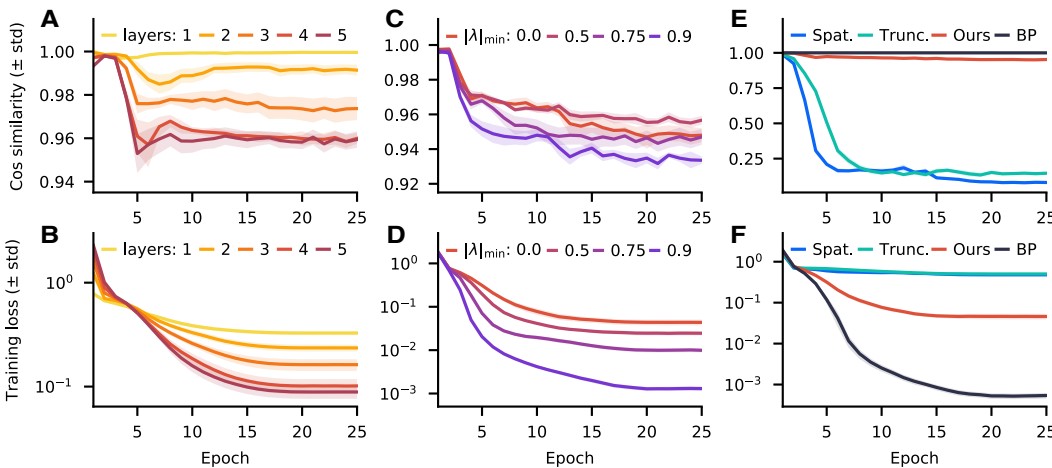

Figure 2: Impact of depth of the network (left), eigenvalues of the network (middle), and type of approximation on the quality of online learning (right). The cosine similarity measures the alignment between the estimated gradient (with our learning rule for all panels, and with spatial backpropagation (Spat.) and 1-step truncated backpropagation (Trunc.) for panels E and F) and the true gradient, computed with backpropagation-through-time (BP). It is computed per layer and then averaged across layers to make a quantitative comparison possible. The encoder and decoder alignments do not influence this metric. This task is solved ($100\%$ accuracy) for losses lower than $0.05$ and $70\%$ accuracy roughly corresponds to a loss of $0.5$. See Section 4.1 for more details.

As remarked in Section 3.2, the updates prescribed by our learning rule match the gradient exactly for all parameters in the last LRU layer. We confirm that empirically for a network of one layer in Figure 2.A. While the approximation quality deteriorates with increasing depth (Fig. 2.A), alignment remains high, noticeably better than for all baseline online learning rules (Fig. 2.E). Moreover, despite alignment decreasing with depth, performance enhances significantly (Fig. 2.B). BPTT exhibits similar improvements with depth, suggesting that this is likely due to the enhanced capabilities of the model. Our rule can learn useful hierarchical temporal representations online whereas baseline methods, 1-step truncated and spatial backpropagation, which ignore most of temporal dependencies, fail (c.f. Fig. 2.F). Additionally, we found that, despite its bias, our learning rule can decrease the loss to 0 when training a 4 layers network on a simpler memory task for long enough.

Exact error terms are backpropagated through recurrent units by weighting the current prediction error by 1 and the future error by $\lambda$. In order to maintain an online algorithm, we ignore this dependency on the future. We expect alignment with the gradient to decrease as the distribution of $|\lambda|$ shifts towards 1. To test that, we initialize the $\lambda$ uniformly at random in the complex ring of radius $[|\lambda|_{\min}, 1]$. Interestingly, alterations in the initial eigenvalue distribution only slightly affect estimation quality in the beginning of training (Fig. 2.C). The key factor seems to be the degradation associated with learning progress, rather than degradation due to larger eigenvalues. Smaller initial $|\lambda|_{\min}$ values slow down training, as more perfect memory neurons have to be recruited, but all initializations eventually lead to solving the task (Fig. 2.D).

## 4.2 Independent recurrent modules improve online learning performance

After dissecting the learning dynamics of our algorithm, we next show the importance of the independence of recurrent modules for online gradient-based learning. To this end, we compare linear recurrent units to densely-connected linear recurrent layers which do not have independent recurrent modules. To make the comparison fair, we ensure all models have the same number of parameters and recurrent neurons. In addition to the baselines we considered in the previous section, we include an extended version of the SnAp-1 algorithm that combines spatially backpropagated errors with the SnAp-1 sensitivity approximation. This algorithm reduces to ours when applied to networks of independent recurrent modules. Therefore, it enables us to isolate the impact of the independent recurrent module design on online gradient-based learning. We report final training losses in Table 1 and refer the reader to Appendix B for experimental details.

Table 1: Comparison of final training losses of different online learning algorithms on the copy task of Section 4.2. The independent recurrent modules design improves online learning performance. Performance greatly degrades when the LRU is replaced with a dense recurrent matrix (Linear RNN). Comparison with the SnAp-1 algorithm applied to the GRU architecture highlights that online learning of multilayer networks is difficult without element-wise recurrence. Results are averaged over 5 seeds.

| Layer
Number layers | LRU
4 | Linear RNN
4 | GRU
1 | GRU
4 |
|---|---|---|---|---|
| SPATIAL | $4.66 \times 10^{-1}$ | $6.20 \times 10^{-1}$ | $6.26 \times 10^{-1}$ | $6.55 \times 10^{-1}$ |
| TRUNCATED | $2.62 \times 10^{-1}$ | $5.81 \times 10^{-1}$ | $6.20 \times 10^{-1}$ | $6.49 \times 10^{-1}$ |
| OURS / SNAP-1 | $8.44 \times 10^{-3}$ | $5.82 \times 10^{-1}$ | $3.16 \times 10^{-1}$ | $3.27 \times 10^{-1}$ |
| BPTT | $7.59 \times 10^{-6}$ | $1.07 \times 10^{-4}$ | $2.61 \times 10^{-1}$ | $1.94 \times 10^{-1}$ |

Table 2: Test accuracy on three tasks of the LRA benchmark [24] for spatial backpropagation, 1-step truncated backpropagation, our algorithm, and full backpropagation through-time. While we always use per time step local losses during training, we accumulate logits over the sequence during inference. We report the mean and std. for three seeds each.

| Method | sCIFAR | IMDB | LISTOPS |
|---|---|---|---|
| SPATIAL | $58.20 \pm 0.70$ | $83.50 \pm 0.20$ | $32.02 \pm 0.27$ |
| TRUNC. | $60.01 \pm 1.26$ | $84.04 \pm 0.47$ | $31.88 \pm 0.59$ |
| OURS | $79.59 \pm 1.01$ | $86.48 \pm 0.41$ | $37.62 \pm 0.68$ |
| BPTT | $83.40 \pm 1.54$ | $87.69 \pm 0.39$ | $39.75 \pm 0.17$ |

Table 3: Test accuracy of a linear RNN on the CIFAR task. Instead of our learning rule, we apply the SnAp-1 learning rule extended to the multilayer case, as described in Section 4.2.

| sCIFAR |
|---|
| $50.63 \pm 0.23$ |
| $50.53 \pm 0.43$ |
| $63.71 \pm 0.33$ |
| $65.23 \pm 0.56$ |

Our findings confirm the benefits of independent recurrent modules for online learning, in particular for multi-layer networks. To demonstrate that, we first compare our algorithm on the LRU architecture with the SnAp-1 algorithm applied to an equivalent linear recurrent network. The diagonal approximation of the sensitivity tensor in SnAp-1 introduces an additional bias when learning linear RNNs. We found that this additional bias hurts performance: when moving from offline BPTT to online training, the performance drop is significantly higher for linear RNNs. Interestingly, sensitivity approximation does not bring any performance gain, in this setting, compared to the cruder approximations that are 1-step truncated BPTT and spatial backpropagation.

Additionally, we run experiments on another RNN architecture, the GRU [44], to better understand the impact of depth in online and offline RNN training, and to confirm the importance of element-wise recurrence for online learning. In the single layer case, consistent with Menick et al. [15], we find that the SnAp-1 approximation performs competitively with offline BPTT. However, it suffers from depth in contrast to BPTT that benefits from it. This result highlights the importance of depth in this memory task, as well as the difficulty learning over depth poses for existing online learning algorithms.

## 4.3 Scaling online learning to the long-range arena benchmark

While the approximations typically employed to enable online learning prohibit scaling to tasks with extended temporal structure, the results from our previous section have demonstrated the potential of independent linear units for online learning of long-range dependencies. We therefore move to tasks from the challenging long-range arena benchmark [24] specifically designed to evaluate this ability. Transformers excel in almost any benchmark today. However, they perform surprisingly subpar in this setting [45] in which deep state-space models [20] and LRUs [23] achieve impressive results.

We run experiments on three tasks, sequential CIFAR, IMDB and LISTOPS. In sequential CIFAR, the network receives the $32 \times 32$ image as a pixel sequence and has to perform a classification task. In line with Orvieto et al. [23], we use the colored version of sCIFAR instead of the grayscale version

originally proposed. In the IMDB task, the network is given a text encode in bytes of length at most 4000, and has to perform binary classification. In LISTOPS, the input is a sequence of numbers, brackets and operators like MAX which the model needs to evaluate to determine a classification target in the range from 1 to 10. We do not use the three other tasks of the LRA benchmark: the performance gap between different models is usually small in the RETRIEVAL task (c.f. [23]) and, in our preliminary experiments, we could not reach above chance performance in the PATHFINDER tasks with BPTT and the modifications we made to make the loss causal, as described in the next paragraph.

In order to make the sequence models employed on this benchmark compatible with the causality requirement in online learning, we remove the time pooling operation during training and consider a local loss term at every time step instead. During inference, we then evaluate our models using the average of the last layer logits in time which respects causality. Moreover, we replace batch normalization with layer normalization to avoid sending batch statistics backwards in time and consider smaller models to lower the computational burden for online learning. For further experimental details, please refer to Appendix B.

We report results comparing online learning to spatial backpropagation, truncated BPTT and the BPTT upper bound in Table 2. Our online learning algorithm outperforms other online learning approximations, significantly reducing the gap towards BPTT. As in the last section, replacing the LRU with a linear RNN layer in the CIFAR experiment leads to worth online learning performance, c.f. Table 3, providing further evidence for the effectiveness of independent recurrent modules for capturing long-term dependencies.

## 5 Discussion

We have demonstrated that long-range dependencies can be learned online, allowing recurrent neural networks to reach strong performance on a set of tasks from the long-range arena benchmark. Moreover, a detailed analysis of a memory problem revealed that our method significantly outperforms both spatial (online) backpropagation as well as prior approaches based on approximate real-time recurrent learning, coming close to full backpropagation-through-time. These findings may inform the design of new neuromorphic hardware with on-chip learning capabilities, an application where approximate real-time recurrent learning is garnering significant attention [46–48].

While most prior related work focused on developing generic gradient approximation schemes, we asked which architecture would simplify online gradient computations. In high-level terms, our philosophy draws from seminal work on long short-term memory networks [LSTMs; 40] or neural Turing machines [49], which established the importance of architecture design for the success of gradient descent. We build on this insight, moving to the harder setting of online learning. This led us to consider networks built of recurrent independent modules: decoupled units with low-dimensional state vectors, for which exact real-time recurrent learning is cheap. Importantly, this design underlies recent models such as deep linear recurrent units [23] and members of the HiPPO family [21, 22] which achieve strong performance in a wide array of challenging problems, including language modeling at scale [50] and the long-range arena benchmark [22].

We conclude by noting that modularity, the overarching principle behind our approach, is at the very heart of the influential columnar hypothesis in neuroscience [51]. This hypothesis states that the architecture of the neocortex is modular, with the cortical column as an elementary (or canonical, [52]) building block one level of abstraction above neurons. We thus speculate that modularity could be a key neural network design principle discovered by evolution, that considerably simplifies the temporal credit assignment problem. This is inline with our finding that a modular architecture enables learning complicated temporal dependencies through simple local temporal credit assignment mechanisms, letting spatial backpropagation take care of assigning credit over the network hierarchy. We stress this point because numerous biological implementations and alternatives for spatial backpropagation have been proposed [e.g., 53–62], while essentially none exist yet for backpropagation-through-time [63]. Our findings provide a starting point for understanding how the brain deals with the fundamental problem of learning the temporal structure behind its sensory inputs.

## Acknowledgments and Disclosure of Funding

All five authors thank Angelika Steger for her support and overarching guidance. João Sacramento thanks Timothy Lillicrap and Greg Wayne for an inspiring early discussion on searching architectures that can learn through limited forms of backpropagation-through-time, and Owen He and Johannes von Oswald for valuable discussions on deep state space models. This research was supported by an Ambizione grant (PZ00P3_186027) from the Swiss National Science Foundation and an ETH Research Grant (ETH-23 21-1). Robert Meier and Asier Mujika were supported by grant no. CRSII5_173721 of the Swiss National Science Foundation.

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

# A  Network architecture and algorithm

## A.1  The linear recurrent unit (LRU)

In Section 2.2, we used a simplified version of the LRU for brevity. Here, we review the more detailed formulation of Orvieto et al. [23]. The LRU is defined as follows:

$$h_{t+1} = \lambda \odot h_t + \gamma \odot Bx_t \tag{8}$$
$$y_t = \mathrm{Re}[Ch_t] + Dx_t. \tag{9}$$

The only difference with Equation 1 is the inclusion of the normalization factor $\gamma \in \mathbb{R}^N$. Its purpose is to ensure that all units maintain a comparable magnitude. Without normalization, hidden states whose $\lambda$ is close to 1 can blow up. We initialize $\gamma$ with

$$\gamma = \sqrt{1 - |\lambda|^2}, \tag{10}$$

and later update it with our learning algorithm.

We use an exponential parametrization for $\lambda$:

$$\lambda := \exp(-\exp(\nu^{\log}) + i\exp(\theta^{\log})). \tag{11}$$

This parametrization ensures that the norm of $\lambda$, equal to $\exp(-\exp(\nu^{\log}))$, remains below 1, guaranteeing stability of the network dynamics. This feature provides an advantage to the LRU compared to the linear RNN, as observed in Section 4.2. By representing $\theta$ as $\exp(\theta^{\log})$, we achieve finer tuning of $\theta$ around 0. In the following, we focus on computing gradients with respect to $\lambda$. However, in our simulations, we derive the update for $\nu^{\log}$ and $\theta^{\log}$ from these gradients through the chain rule, and apply them to update the corresponding parameters. Similarly, we optimize the logarithm of $\gamma$.

## A.2  Complete derivation of our algorithm

We provide a more detailed derivation of our online learning rule for a single layer of LRUs, as described in Section 3.1. Recall that we define $e_t$ to be the non-zero terms of the sensitivity $\mathrm{d}_\theta h_t$ of the state $h_t$ with respect to the parameters $\theta$, with $\theta = \{\lambda, \gamma, B\}$ the parameters of the LRU impacting the recurrent dynamics. We show that the sensitivities evolve according to

$$e_{t+1}^\lambda = \lambda \odot e_t^\lambda + h_t \tag{12}$$
$$e_{t+1}^\gamma = \lambda \odot e_t^\gamma + Bx_{t+1} \tag{13}$$
$$e_{t+1}^B = \mathrm{diag}(\lambda)e_t^B + \gamma x_{t+1}^\top \tag{14}$$

and that gradient-following parameter updates can be calculated through

$$\Delta\lambda \propto \sum_{t=1}^T \delta_t \odot e_t^\lambda \tag{15}$$

$$\Delta\gamma \propto \sum_{t=1}^T \mathrm{Re}[\delta_t \odot e_t^\gamma] \tag{16}$$

$$\Delta B \propto \sum_{t=1}^T \mathrm{diag}(\delta_t)e_t^B, \tag{17}$$

with $\delta_t = \mathrm{d}_{h_t}L_t$. These updates are the ones used in our simulations. We now derive the updates per coordinate, and the vectorized form can be straightforwardly obtained from there.

**Update for $\lambda$.** We have

$$e_{t+1,i}^\lambda = \frac{\mathrm{d}h_{t+1,i}}{\mathrm{d}\lambda_i} = \frac{\mathrm{d}}{\mathrm{d}\lambda_i}[\lambda_i h_{t,i}] + 0 = \lambda_i e_{t,i}^\lambda + h_{t,i}. \tag{18}$$

The first equality uses the definition of $e_{t+1}^\lambda$, the second the recurrent update of $h_t$ as in Equation 8 and the fact that $\gamma \odot Bx_{t+1}$ does not depend on $\lambda$, and the third is the complex product rule. The

gradient of the loss with respect to $\lambda_i$ is then equal to

$$\frac{\mathrm{d}L}{\mathrm{d}\lambda_i} = \sum_{t=1}^{T} \sum_{j} \frac{\mathrm{d}L_t}{\mathrm{d}h_{t,j}} \frac{\mathrm{d}h_{t,j}}{\mathrm{d}\lambda_i} \tag{19}$$

$$= \sum_{t=1}^{T} \frac{\mathrm{d}L_t}{\mathrm{d}h_{t,i}} \frac{\mathrm{d}h_{t,i}}{\mathrm{d}\lambda_i} \tag{20}$$

$$= \sum_{t=1}^{T} \delta_{t,i} e_{t,i}^{\lambda}. \tag{21}$$

In the first line, we applied forward-mode complex differentiation (combined with $h_t$ being an holomorphic function of $\lambda$) as in Equation 5. In the second, we used that $h_i$ is independent of $\lambda_j$ for $i \neq j$.

**Update for $\gamma$.** The derivation for the recurrent update of $e^{\gamma}$ being very similar to the one for $\lambda$, we omit it. The fact that $\gamma$ is a real variable does not change anything to this derivation. However, this makes the gradient computation slightly different. One can act as if $\gamma$ was a complex variable $\gamma_i^{\mathbb{C}}$ and get

$$\frac{\mathrm{d}L}{\mathrm{d}\gamma_i^{\mathbb{C}}} = \sum_{t=1}^{T} \delta_{t,i} e_{t,i}^{\gamma}. \tag{22}$$

Now, looking at the real part of the previous equation, we have

$$\frac{\mathrm{d}L}{\mathrm{d}\gamma_i} = 2 \sum_{t=1}^{T} \mathrm{Re}[\delta_{t,i} e_{t,i}^{\gamma}]. \tag{23}$$

Note that the factor 2 will also indirectly appear in the update of $\lambda$, although it is not yet there in the gradient, as we have a real derivative here, when it is a complex one above.

**Update for $B$.** We have

$$e_{t+1,ji}^{B} = \frac{\mathrm{d}h_{t+1,j}}{\mathrm{d}B_{ji}} = \frac{\mathrm{d}}{\mathrm{d}B_{ji}} [\lambda_j h_{t,j} + \gamma_j B_{ji} x_{t+1,i}] = \lambda_j \frac{\mathrm{d}h_{t,i}}{\mathrm{d}B_{ji}} + \gamma_j x_{t+1,i} \tag{24}$$

and

$$\frac{\mathrm{d}L}{\mathrm{d}B_{ji}} = \sum_{t=1}^{T} \delta_{t,j} e_{t,ji}^{B}. \tag{25}$$

Note that the update for $e^B$ is more general than the one of Equation 6 in the main text in which $\gamma = 1$.

## B  Experimental details

We base our implementation on the S5 [22] code base[4]. All networks are trained with the AdamW optimizer, with a linear learning rate warm up, followed by a one-cycle cosine learning rate decay. Following common practice, we do not apply weight decay for $\lambda$, $\gamma$, and use a smaller learning rate for those parameters (global learning rate times a learning rate factor). By default, we initialize $\lambda$ uniformly at random in the complex disk and $\theta$ uniformly at random in $[0, 2\pi]$.

### B.1  Copy task experimental details and hyperparameters

We report the hyperparameters we used and scanned over for the copy task in Tables 4 and 5. In Section 4.1, we take the default configuration reported in the LRU paper [23] for backpropagation-through-time and apply it to the different methods we consider. We use 25 epochs, with a linear warmup of 5 epochs. We tuned the learning rate for each method independently in the comparison of Figure 2.E and F and in the one of Table 1. For the 1-layer GRU architecture, following Menick et al. [15], we add an extra readout layer: a 1-hidden layer MLP with hidden dimension 1072 processes the hidden state to generate the output.

---

[4] https://github.com/lindermanlab/S5

Table 4: Hyperparameter configurations for Section 4.1. We use $[\cdots]$ to denote hyperparameters that were scanned over with grid search and $\{\cdots\}$ to denote the variables of interest for the figure.

| Hyperparameter | FIG.2.A/B | FIG.2.C/D | FIG.2.E/F |
|---|---|---|---|
| Learning rule | Ours | Ours | {Ours, Spat., Trunc., BP} |
| Pattern length | 20 | 20 | 20 |
| Padding | 7 | 7 | 7 |
| Training samples | 20000 | 20000 | 20000 |
| Number of layers | $\{1, 2, 3, 4\}$ | 4 | 4 |
| Recurrent state size $N$ | 64 | 64 | 64 |
| Model size $H$ | 128 | 128 | 128 |
| $\|\lambda_{\min}\|$ | 0 | $\{0, 0.5, 0.75, 0.9\}$ | 0 |
| Epochs | 25 | 25 | 25 |
| Warmup | 0 | 0 | 0 |
| Batch-size | 50 | 50 | 50 |
| Base learning rate | $10^{-3}$ | $2 \times 10^{-3}$ | $2^{[0,1,2,3]}10^{-3}$ |
| Learning rate factor | 0.5 | 0.5 | 0.5 |
| Dropout probability | 0.1 | 0.1 | 0.1 |
| Weight-decay | 0 | 0 | 0 |

Table 5: Hyperparameter configurations for Section 4.2. We use $[\cdots]$ to denote hyperparameters that were scanned over.

| Layer | LRU | Linear RNN | GRU | GRU |
|---|---|---|---|---|
| Number layers | 4 | 4 | 1 | 4 |
| Pattern length | 20 | 20 | 20 | 20 |
| Padding | 7 | 7 | 7 | 7 |
| Training samples | 20000 | 20000 | 20000 | 20000 |
| GLU | Yes | Yes | No | Yes |
| $N$ | 64 | 64 | 134 | 91 |
| $H$ | 128 | 146 | 134 | 91 |
| Extra readout | No | No | 1072 | No |
| Number parameters | 268,430 | 267,956 | 269,488 | 270,011 |
| Batch-size | 20 | 20 | 20 | 20 |
| Base learning rate | $2^{[0,\cdots,5]}10^{-3}$ | $2^{[0,\cdots,5]}10^{-3}$ | $2^{[0,\cdots,5]}10^{-3}$ | $2^{[0,\cdots,5]}10^{-3}$ |
| Learning rate factor | 1 | 1 | 1 | 1 |
| Dropout probability | 0.1 | 0.1 | 0.1 | 0.1 |
| Weight-decay | 0 | 0 | 0 | 0 |

Table 6: Hyperparameter configurations for SCIFAR, IMDB and LISTOPS experiments. We use $[\cdots]$ to denote hyperparameters that were scanned over with grid search. By default, the $\{\lambda, \gamma\}$ parameters have no weight decay and have a slower learning rate (c.f. learning rate factor).

| Hyperparameter | CIFAR | IMDB | LISTOPS |
|---|---|---|---|
| Number of layers | 4 | 4 | 4 |
| Recurrent state size $N$ | 128 | 128 | 128 |
| Model size $H$ | 256 | 256 | 256 |
| $\|\lambda\|_{\min}$ | 0.9 | $[0.0, 0.9]$ | $[0.0, 0.9]$ |
| $\|\lambda\|_{\max}$ | 0.999 | 1 | 1 |
| Batch-size | 100 | 32 | 32 |
| Base learning rate | $[0.001, 0.004]$ | $[0.001, 0.003]$ | $[0.001, 0.003]$ |
| Learning rate factor | 0.5 | 0.5 | 0.5 |
| Dropout probability | 0.1 | 0 | 0.1 |
| Weight-decay | $[0.1, 0.5]$ | 0.05 | 0.05 |
| Epochs | 180 | 40 | 35 |
| Warmup | 18 | 4 | 0 |

## B.2 Experimental details and hyperparameters for SCIFAR, IMDB and LISTOPS

For all experiments, we first ran a manual coarse-grained hyperparameter tuning to identify the most important parameters, and then ran the grid search described in 6. The final hyperparameters were each evaluated on 3 fresh seeds for the results reported in Table 2. The training time for our online learning rule on a single Nvidia RTX3090 GPU for SCIFAR, IMDB and LISTOPS was respectively 36, 10 and 40 hours.

For the comparison with linear RNNs, we kept the number $N$ of hidden neurons fixed compared to the one used the LRU (c.f. Table 6), but changed the model size $H$ to 294 in order to obtain the same number of parameters (our Linear RNN has 1,068,086 parameters vs. 1,063,178 for the LRU). We used the same hyperparameter optimization scheme.

