# OpenReview forum: "Online learning of long-range dependencies"
_NeurIPS.cc/2023/Conference — NeurIPS 2023 poster_

### Official Review · Reviewer_hL6u · 2023-07-04

**Soundness:** 3 good
**Presentation:** 3 good
**Contribution:** 3 good
**Rating:** 7
**Confidence:** 4

**Summary:**

The paper looks at the problem of online learning in RNNs from a perspective of exact gradient computation: it looks for network architectures for which exact gradients can be computed. It uses the recently proposed LRU (Orvieto et. al., 2023) to show that with linear and diagonal structure in recurrent dependencies, one can tractably compute the exact gradients online for a single layer network. For multi-layer networks, this simplified structure combined with truncation of gradient signal (only considering current time step) results in an approximate gradient.

The paper performs two sets of experiments. The first set examines the difference in the approximate gradient computed using the proposed method with the exact gradient. The second set of experiments focuses on learning tasks that require capturing long range dependencies. In both set of experiments the proposed method closes the gap to BPTT compared to the baselines used in the paper.

**Strengths:**

### Originality

The paper combines two known ideas, independent linear recurrent modules (LRU from Orvieto et. al., 2023) and gradient sparsification for online learning (Menick et. al., 2021), and presents them in new light.

### Quality
1. The paper is technically sound, has carefully designed experiments that support the claims well.
2. claims supported by theory and/or experiments
### Clarity
1. The paper is well organized and clear for most part. There are some parts that can be improved (see weaknesses).
### Significance
1. The well designed experiments make the results presented in the paper useful for the community.

**Weaknesses:**

### Quality
1. Along with other claims that are well supported, the authors claim that they have pushed the standard for what is possible through online recurrent learning. However, I believe that the current experiments, which are limited to simple long range dependency tasks, are not sufficient for such a grand claim. Specifically, since all the experiments in the paper use synthetic data, it will be nice, for example, to include an experiment on real world data like small scale language modeling and compare the results to an LSTM language model, and discuss the limitations of the work in terms of performance on real world data that has complex dependency structure that is not simple long-range structure. Such an experiment will show the limitations of online learning and help guide further research on the topic.
2. Why does table 1 not contain sparse approximation method like SnAp-n applied to dense RNNs?

### Clarity

Following  are some mistakes/typos in the paper that can throw a reader off.
1. There is a mix-up in the symbols in equation 1, and line 103. I think $x_t$ should be replaced with $h_t$ in Eq 1, and line 103 should state that $u_t$ is the input at time $t$.
2. Line 109-111: I think you mean to say *absence of temporal non-linearity* in line 110.

Following are some suggestions that could improve the paper:
1. While the authors provide through discussion for single-layer networks (Sec 3.1), the more important case of multi-layer networks does not get the same amount of care. The entire discussion in section 3.2 happens without the use of detailed expressions. Figure 1 does provide some support to the discussion but I find it to be insufficient. At the least, the authors should accompany the pictorial description in Fig 1 with expression for $\delta$ at each layer. This will help with the discussion of various points that come up in the experiment section 4.1 that centers around effect of dept. Specifically, an expression for $\delta$ at a lower layer it will help the point made in line 201-205, that is later discussed in section 4.1 around line 245.

**Questions:**

### Typos and suggestions
1. Line 130: Using them for complex differentiation allows using calculus rules that are similar to those for real valued functions.
2. Line 167: "Those" as a vague reference here might cause confusion. It might be better to say $e^\lambda$ and $e^B$ explicitly.

**Limitations:**

Along with other claims that are well supported, the authors claim that they have pushed the standard for what is possible through online recurrent learning. However, I believe that the current experiments, which are limited to simple long range dependency tasks, are not sufficient for such a grand claim. Specifically, since all the experiments in the paper use synthetic data, it will be nice, for example, to include an experiment on real world data like small scale language modeling and compare the results to an LSTM language model, and discuss the limitations of the work in terms of performance on real world data that has complex dependency structure that is not simple long-range structure. Such an experiment will show the limitations of online learning and help guide further research on the topic.

---

> ### Author Rebuttal · Authors · 2023-08-09
>
> Thank you for your useful comments and thoughts. We reply below to each point raised individually.
>
> > Along with other claims that are well supported, the authors claim that they have pushed the standard for what is possible through online recurrent learning. However, I believe that the current experiments, which are limited to simple long range dependency tasks, are not sufficient for such a grand claim. Specifically, since all the experiments in the paper use synthetic data, it will be nice, for example, to include an experiment on real world data like small scale language modeling and compare the results to an LSTM language model, and discuss the limitations of the work in terms of performance on real world data that has complex dependency structure that is not simple long-range structure. Such an experiment will show the limitations of online learning and help guide further research on the topic.
>
> We would like to note that approximate RTRL research is still in a proof-of-concept phase, and prior work has only considered at best only very small-scale language modeling experiments. These would be the experiments within our reach. The problem with this is that it seems likely that such small-scale experiments would not reveal the ability to model long-range dependencies; our impression based on prior work is that most of the capacity is used to learn dominant short-range dependencies with small RNN models. This may be one of the reasons why the SnAp authors found surprisingly good results on small-scale character-level modeling already with reservoir (not learned) GRU networks, and why SnAp almost matched BPTT. We would also note that the LRA is not entirely toyish, and a challenging problem suite; for example, transformer models are not strong performers there, and they require large attention spans on the order of hundreds of tokens [24].
>
> We have however experimented with single- and multilayer GRU networks on the copy task (cf. [global response](https://openreview.net/forum?id=Wa1GGPqjUn&noteId=HB3E5CrCGS)), trained with approximate RTRL (SnAp combined with spatial backpropagation to allow training deep networks, as we did for our models), BPTT, TBPTT and spatial backpropagation, and found that such networks are greatly outperformed by LRU networks when trained online. Interestingly, only BPTT can leverage the additional capacity of deep GRU networks, whereas online approximate RTRL stagnates at some value corresponding to a 1-layer GRU. This again shows the power of approximate online RTRL when paired with networks with independent recurrent modules. We are currently running experiments with both single- and multilayer GRU networks on sequential CIFAR and we will add those results to the next version of the paper.
>
> > Why does table 1 not contain sparse approximation method like SnAp-n applied to dense RNNs?
>
> We now trained dense linear RNNs using BPTT, truncated BPTT, spatial backpropagation and our hybrid learning rule (which combines SnAp-like forward sensitivity propagation with spatial backpropagation) on sequential CIFAR, cf. [global response](https://openreview.net/forum?id=Wa1GGPqjUn&noteId=HB3E5CrCGS) results pdf. Our online-learned LRUs greatly outperform online-learned dense linear RNNs. The next version of the paper will include dense linear RNN results for the remaining LRA tasks considered here, ListOps and IMDB.
>
> > While the authors provide through discussion for single-layer networks (Sec 3.1), the more important case of multi-layer networks does not get the same amount of care. The entire discussion in section 3.2 happens without the use of detailed expressions. Figure 1 does provide some support to the discussion but I find it to be insufficient. At the least, the authors should accompany the pictorial description in Fig 1 with expression for $\delta$ at each layer. This will help with the discussion of various points that come up in the experiment section 4.1 that centers around effect of dept. Specifically, an expression for $\delta$ at a lower layer it will help the point made in line 201-205, that is later discussed in section 4.1 around line 245.
>
> Thank you for this suggestion with which we agree. We will expand section 3.2 with expressions for the spatially backpropagated error $\delta$ and improve Fig. 1 following your suggestion.
>
> > There is a mix-up in the symbols in equation 1, and line 103. I think $x_t$ should be replaced with $h_t$ in Eq 1, and line 103 should state that $u_t$ is the input at time $t$.
> > Line 109-111: I think you mean to say absence of temporal non-linearity in line 110.Line 130: Using them for complex differentiation allows using calculus rules that are similar to those for real valued functions.Line 167: "Those" as a vague reference here might cause confusion. It might be better to say $e^\lambda$ and $e^\gamma$ explicitly.
>
> Thank you for catching these mistakes, which we will correct in the next version of the paper.
>
> > Along with other claims that are well supported, the authors claim that they have pushed the standard for what is possible through online recurrent learning. However, I believe that the current experiments, which are limited to simple long range dependency tasks, are not sufficient for such a grand claim.
>
> We did not wish to oversell our claims. We will downtone our discussion presenting our results as promising while clarifying that they are still limited to relatively small-scale image (sequential CIFAR), text (IMDB) classification and symbol manipulation (ListOps) tasks.
>
> We remain fully available to respond to any further questions you may have during the discussion period.

---

> > ### Comment · Reviewer_hL6u · 2023-08-15
> > **Thank you for the thorough response**
> >
> > I thank the authors for their thorough response and for providing additional results. With the additional information, I now think that the paper can have significant impact, at least on RTRL community. I've increased my score by one point to reflect this.

---

### Official Review · Reviewer_53aD · 2023-07-05

**Soundness:** 3 good
**Presentation:** 2 fair
**Contribution:** 3 good
**Rating:** 6
**Confidence:** 3

**Summary:**

The authors provide an online learning algorithm for linear recurrent units [23]. They take advantage of the fact that each unit of the LRU is an ‘independent recurrent module’ and thus RTRL for each LRU layer simplifies substantially in this case and becomes tractable. They test on some long-range dependency tasks.

**Strengths:**

-	The authors have combined insights from recent developments like LRUs being as good as RNNs while being easier to train, and approximations of RTRL for online local learning as in biology.
-	They demonstrate good results on some long-range dependency tasks as compared to RNNs.


**Weaknesses:**

-	The learning algorithm proposed seems to me just e-prop [13] applied to the LRU. Indeed, e-prop also takes into account self-recurrence of each unit and 1-step lateral recurrence. With LRU, since there is no lateral recurrence, e-prop becomes exact within one LRU layer.
-	Their algorithm still relies on spatial backprop between layers, so this is still not local in space.
-	The authors claim that “numerous biological implementations and alternatives for spatial backpropagation have been proposed [e.g., 50–59], while essentially none exist yet for backpropagation-through-time [60].”. But various approximations for BPTT have also been proposed like [13].


**Questions:**

-	The authors should study the relation with e-prop and inform if there are any differences with it. To me, their algorithm is the same.
-	In Fig. 3, I expect that BP is BPTT, and Spat. Is just the spatial BPTT without temporal dependencies? If not, was BPTT not used?

Minor:
-	Mistakes in equation (1)! Cf. appendix A. I’ve classified this as minor, but it does not inspire confidence that the very first equation has many mistakes.
-	L304: inference -> inference


**Limitations:**

N/A.

---

> ### Author Rebuttal · Authors · 2023-08-09
>
> Thank you for your useful review. We reply to your specific questions and concerns individually below.
>
> > The learning algorithm proposed seems to me just e-prop [13] applied to the LRU. Indeed, e-prop also takes into account self-recurrence of each unit and 1-step lateral recurrence. With LRU, since there is no lateral recurrence, e-prop becomes exact within one LRU layer. [...] The authors should study the relation with e-prop and inform if there are any differences with it. To me, their algorithm is the same.
>
> Our learning rule is indeed very closely related to e-prop and SnAp. We will update our manuscript with a detailed discussion, cf. [global response](https://openreview.net/forum?id=Wa1GGPqjUn&noteId=HB3E5CrCGS). Besides being able to learn complex neurons, our rule handles multilayer recurrent neural networks (RNNs) differently, something that we failed to point out previously. More concretely, e-prop and SnAp-1 do not send detailed spatial credit assignment information in deep RNN models such as the LRU. At best it only broadcasts a global error backwards through skip connections, and in the worst case for networks without skip connections it leads to identically zero updates to deep hidden recurrent neurons (that do not directly influence the loss). By combining forward sensitivity propagation with spatially backpropagated errors, our hybrid learning rule delivers detailed learning signals even to hidden recurrent neurons, while retaining the cost of a single inference pass.
>
> > Their algorithm still relies on spatial backprop between layers, so this is still not local in space.
>
> Yes. In this manuscript, we opted to present the architecture and algorithm in its purest form and attack the challenging LRA suite. We will add a comment to the next version of the paper, explicitly suggesting that it would be interesting to explore replacing spatial backpropagation by approximations such as feedback alignment (Lillicrap et al., 2015) in future work.
>
> > The authors claim that “numerous biological implementations and alternatives for spatial backpropagation have been proposed [e.g., 50–59], while essentially none exist yet for backpropagation-through-time [60].”. But various approximations for BPTT have also been proposed like [13].
>
> We originally intended to emphasize that the rule only needs spatial backpropagation and not reverse-mode temporal backpropagation, in the strict sense (not encompassing forward-mode differentiation), but the remark ended up being confusing. We will remove it from the discussion in the next version of the paper.
>
> > In Fig. 3, I expect that BP is BPTT, and Spat. Is just the spatial BPTT without temporal dependencies? If not, was BPTT not used?
>
> Yes, BP is BPTT, and Spat. is spatial backpropagation. This will be clarified.
>
> > Minor: - Mistakes in equation (1)! Cf. appendix A. I’ve classified this as minor, but it does not inspire confidence that the very first equation has many mistakes.L304: inference -> inference
>
> Thank you for spotting these typos, which will be corrected in the next version of the paper.
>
> We are more than happy to answer additional questions that you may have during the discussion period. We note that we ran additional experiments triggered by other reviews, that we collected in the [global response](https://openreview.net/forum?id=Wa1GGPqjUn&noteId=HB3E5CrCGS).

---

> > ### Comment · Reviewer_53aD · 2023-08-15
> > **read rebuttal**
> >
> > I have read the authors' rebuttal. They have accepted the issues pointed out by others and me and agree to elaborate on / clarify these issues. I maintain my rating of  6: weak accept.
> >
> > Minor point: Instead of feedback alignment (Lillicrap et al 2015) for local spatial backprop, look at Akrout et al 2019 and the earlier references therein -- far before ML discovered feedback alignment with some fanfare, computational neuroscientists were just learning the feedback weights to align correctly which works much better than feedback alignment!

---

### Official Review · Reviewer_14Gy · 2023-07-06

**Soundness:** 3 good
**Presentation:** 4 excellent
**Contribution:** 2 fair
**Rating:** 6
**Confidence:** 4

**Summary:**

The authors show that applying an online learning algorithm to independent recurrent modules of linear recurrent units, drastically reduces the algorithm’s computational and memory requirements. They then show numerically that the algorithm’s gradient approximation for multi-layer networks is close to the “real gradient” (as provided by BPTT) and that the algorithm performs well across a range of tasks despite the approximation and using decoupled recurrent units.

**Strengths:**

The authors take a promising path towards effective and efficient gradient-based online learning of recurrent neural networks: Instead of deriving a new approximation of BPTT or RTRL, they choose a network architecture that drastically reduces the computational and memory requirements when deriving online gradient updates for them. The paper provides a comprehensive overview and discusses relations and differences to previous work. The authors compare their proposed network architecture to other architectures and learning algorithms on the copy task. They further evaluate their architecture for three different learning algorithms on the sCIFAR and ListOps task. The paper is very well structured, written and accessible. The algorithm is derived in great detail, including a very helpful primer on complex differentiation, which is very helpful for understanding the research question, results and potential limitations. The figures are generally clear and accessible.

**Weaknesses:**

- While I want to strongly emphasise that I think it is awesome that the authors flag that the SnAp-1 algorithm is reducing to the proposed algorithm when being applied to the proposed network architecture; this opens questions about the contributions made by the paper. In how far is the proposed algorithm new? How is it different from SnAp-1? Maybe a list of contributions at the beginning of the paper could bring clarity? I also would like to flag here that I am not familiar with SnAp-1.
- The authors write that the “most remarkable result” is that their online learning algorithm significantly outperforms learning of networks with densely connected recurrent neurons on the copy task. Given that the copy task requires to memorise a sequence of patterns with i.i.d. sampled entries, is it really surprising that having no interference between hidden units is helpful?
- To show that the decoupled network architecture is performing reliably and well and is a promising alternative to coupled RNNs, in my opinion a comparisons on the more complex tasks, i.e., sCIFAR and ListOps, are missing (i.e. like the data in figure 3) .

Minor:
- Two typos in equation 1 (formula for yt)
- No description for panels A-D in caption of figure 2
- Maybe adding h-lines for 100% and 70% accuracies in figure 2?

**Questions:**

- Where is the high variance of the proposed algorithm in the ListOps task coming from?
- In order to make an online task out of sCIFAR, the authors provide the label at every timestep. How is target encoded / scaled over time? What is the influence on performance of providing the target at each time step?
- In lines 61-62 the authors write “Finally, in Section 4, we analyse our algorithm and relevant baselines [...] with sequence lengths up to over 1000 steps” – what experiment is this referring to and how are 1000 steps required?

**Limitations:**

A discussion of limitations of the work is missing.

---

> ### Author Rebuttal · Authors · 2023-08-09
>
> Thank you for your useful review. We reply to your specific questions and concerns individually below.
>
> > While I want to strongly emphasise that I think it is awesome that the authors flag that the SnAp-1 algorithm is reducing to the proposed algorithm when being applied to the proposed network architecture; this opens questions about the contributions made by the paper. In how far is the proposed algorithm new? How is it different from SnAp-1? Maybe a list of contributions at the beginning of the paper could bring clarity? I also would like to flag here that I am not familiar with SnAp-1.
>
> Thank you for raising this question, which made us realize that we were not precise enough in our comparison with SnAp-1. We will update our manuscript with a detailed discussion, cf. [global response](https://openreview.net/forum?id=Wa1GGPqjUn&noteId=HB3E5CrCGS). Besides being able to learn complex neurons, our rule handles multilayer recurrent neural networks (RNNs) differently, something that we failed to point out previously. More concretely, SnAp-1 does not send detailed spatial credit assignment information in deep RNN models such as the LRU. At best it only broadcasts a global error backwards through skip connections, and in the worst case for networks without skip connections it leads to identically zero updates to deep hidden recurrent neurons (that do not directly influence the loss). Getting a learning signal for such neurons requires raising $k$ in SnAp-$k$, which comes at additional computational and memory costs. In contrast to SnAp, by combining forward sensitivity propagation with spatially backpropagated errors, our hybrid learning rule delivers detailed learning signals even to hidden recurrent neurons, while retaining the cost of a single inference pass.
>
> > The authors write that the “most remarkable result” is that their online learning algorithm significantly outperforms learning of networks with densely connected recurrent neurons on the copy task. Given that the copy task requires to memorise a sequence of patterns with i.i.d. sampled entries, is it really surprising that having no interference between hidden units is helpful?
> >
>
> The independent recurrent module design is indeed well-suited for the copy task. However, we would like to emphasize that (1) our learning rule is still much better than truncated and spatial backpropagation applied to the same architecture (cf. Fig. 2 E/F), and (2) for this task and architecture (at this width), performance still improves with depth (cf. Fig. 2 A/B) when training with our learning rule. Together these two points show that our rule can do deep spatiotemporal credit assignment.
>
> > To show that the decoupled network architecture is performing reliably and well and is a promising alternative to coupled RNNs, in my opinion a comparisons on the more complex tasks, i.e., sCIFAR and ListOps, are missing (i.e. like the data in figure 3) .
> >
>
> We now trained dense linear RNNs using BPTT, truncated BPTT, spatial backpropagation and our hybrid learning rule (which combines SnAp-like forward sensitivity propagation with spatial backpropagation) on sequential CIFAR, cf. [global response](https://openreview.net/forum?id=Wa1GGPqjUn&noteId=HB3E5CrCGS) results pdf). Our online-learned LRUs greatly outperform online-learned dense linear RNNs. The next version of the paper will include dense linear RNN results for the remaining LRA tasks considered here, ListOps and IMDB.
>
> > Minor:Two typos in equation 1 (formula for yt)No description for panels A-D in caption of figure 2Maybe adding h-lines for 100% and 70% accuracies in figure 2?
>
> Thank you for catching the typos and for the helpful figure suggestions that we will take in.
>
> > Where is the high variance of the proposed algorithm in the ListOps task coming from?
>
> After our bug fix (cf. [global response](https://openreview.net/forum?id=Wa1GGPqjUn&noteId=HB3E5CrCGS)), variance is now much smaller (0.68% for our learning rule, vs. 0.17% for BPTT, 0.59% for truncated BPTT and 0.27% for spatial BP).
>
> > In order to make an online task out of sCIFAR, the authors provide the label at every timestep. How is target encoded / scaled over time? What is the influence on performance of providing the target at each time step?
> >
>
> The one-hot encoded target is given at every time step and the total loss is the average of all instantaneous losses. We haven’t run any experiments in which targets are given more rarely.
>
> > In lines 61-62 the authors write “Finally, in Section 4, we analyse our algorithm and relevant baselines [...] with sequence lengths up to over 1000 steps” – what experiment is this referring to and how are 1000 steps required?
> >
>
> The lengths of the sequences we consider are 1024 in sCIFAR, 2048 in ListOps, and 4096 in IMDB.
>
> We hope that these clarifications together with the additional results triggered by other reviewers (cf. [global response](https://openreview.net/forum?id=Wa1GGPqjUn&noteId=HB3E5CrCGS)) help you see our work in a more positive light. We are very happy to address any further questions that may arise during the discussion period.

---

> > ### Comment · Reviewer_14Gy · 2023-08-15
> >
> > I would like to thank the authors for their thorough reply.
> >
> > In acknowledgement of the added comparisons to other algorithms on the sCIFAR, IMDB and ListOPS datasets, the overall quality of the work and presentation - but still wanting to acknowledging the, in my humble opinion, "only" moderate-to-high impact of the work, I have now revised my overall rating to a 6.

---

### Official Review · Reviewer_XofY · 2023-07-07

**Soundness:** 3 good
**Presentation:** 3 good
**Contribution:** 3 good
**Rating:** 6
**Confidence:** 4

**Summary:**

The authors introduce a forward-only approach to gradient computation in linear recurrent layers. The approach yields exact gradients for a single layer and approximate gradients for multilayer networks. They show that their approach yields more accurate gradients and results in more successful learning than alternative approximations that enable forward-only gradient computation in recurrent layers, and they illustrate how their method is particularly adapted to the structure of linear recurrent layers. They show that their approach yields test performance similar to that of backpropagation through time for a multilayer LRU-based model on two tasks in the long-range arena (LRA) benchmark.


**Strengths:**

The context, relevant background, and main contribution are presented clearly. The contribution itself is technically insightful and leverages a view specific to linear, complex-valued recurrent units. The topic of efficient learning in RNNs is well-motivated and contributes to a growing literature on linear recurrent parameterizations for sequence models. The experiments, particularly the illustrations on the copy task, are well-documented and show an insightful framing of both the performance advantages and the anticipated challenges of the proposed method.


**Weaknesses:**

- While the illustration of the method on the method on the copy task is informative, the broader empirical evaluation is limited. Apart from the copy task results, the entire basis for the claim that the algorithmic contribution enables learning of long-range dependencies is a small set of results on a subset of the LRA benchmark. The authors do not disclose why only a subset of LRA is used. The results are difficult to contextualize as it’s not clear what values indicate successful learning of the long-range dependencies in the data. Other comparative evaluations, e.g. in terms of computational or memory costs, are absent. This contribution would be impactful even without state-of-the-art performance along any of these dimensions, but a lack of context is harder to overcome.

- The paper relies heavily on the LRU of Orvieto et al. (2023) and appears to contain some overlap in content with that work (e.g. Fig 1 left is nearly identical to Fig 1 left in Orvieto et al.). At some points, for example $\S$4.2, the evaluation and analysis appears to be at least as much about the LRU itself as the algorithmic contribution of this submission. The authors could improve this work by more clearly delineating the present contribution from that of Orvieto et al. (2023).

- The authors spend half of the discussion arguing for the potential importance of their work for neuroscience, which seems somewhat implausible and at odds with a relatively tight architectural and algorithmic focus up to that point (save for a brief aside on lines 170-171). The similarities in terms of locality, complex values, and modularity seem mostly superficial in light of the larger differences separating forward-only learning of LRUs from computational models of spiking neurons, to say nothing of actual, biological networks of neurons. If this analogy is important, it deserves more careful development in the main body of the paper.


**Questions:**

- Can the authors clarify the evidence for learning of long-range dependence and offer some additional context in line with the first point above? Do *any* of the results in Table 1 suggest that the model has learned long-range dependencies? Can the authors share their reasons for limiting evaluation to a subset of the LRA benchmark?

- What is the relevance of the two paragraphs spanning lines 270-287 with respect to the contribution of the present paper? They seem to be largely focused on an (important) detail for initialization of linear RNN weights. Could this finding be interpreted as evidence that the comparisons are performed suboptimally, and other initialization schemes should have been explored?

- Typically, “online” learning refers to contexts in which data continuously streams from a source and low-cost updates are made in real time, after which an observation is not revisited. Do the experiments here follow this setup, or are multiple forward passes over the data required? If the latter, what is the advantage in practice of this method over BPTT?

**Limitations:**

There is no direct potential for negative social impact. Potential limitations for the method and results are either discussed or covered above.

---

> ### Author Rebuttal · Authors · 2023-08-09
>
> Thank you for the constructive and useful criticism. We reply point by point below.
>
> > The authors spend half of the discussion arguing for the potential importance of their work for neuroscience, which seems somewhat implausible and at odds with a relatively tight architectural and algorithmic focus up to that point (save for a brief aside on lines 170-171). The similarities in terms of locality, complex values, and modularity seem mostly superficial in light of the larger differences separating forward-only learning of LRUs from computational models of spiking neurons, to say nothing of actual, biological networks of neurons. If this analogy is important, it deserves more careful development in the main body of the paper.
>
> While we find our results promising as a starting point for more biologically realistic models, we agree that the discussion balance pended too much towards neuroscience. We will keep our neuroscience discussion to a single paragraph in the next version of the paper, which will read:
> *"We conclude by noting that modularity, the overarching principle behind our approach, is at the very heart of the influential columnar hypothesis in neuroscience (Mountcastle, 1957). This hypothesis states that the architecture of the neocortex is modular, with the cortical column as an elementary (or canonical, cf. Douglas et al., 1989) building block one level of abstraction above neurons. We thus speculate that modularity could be a key neural network design principle discovered by evolution, that considerably simplifies the temporal credit assignment problem. This is in line with our finding that a modular architecture enables learning complicated temporal dependencies through simple local temporal credit assignment mechanisms, letting spatial backpropagation take care of assigning credit over the network hierarchy. Our findings provide a starting point for understanding how the brain deals with the fundamental problem of learning the temporal structure behind its sensory inputs."*
>
> > What is the relevance of the two paragraphs spanning lines 270-287 with respect to the contribution of the present paper? They seem to be largely focused on an (important) detail for initialization of linear RNN weights. Could this finding be interpreted as evidence that the comparisons are performed suboptimally, and other initialization schemes should have been explored?
>
> We also agree that too much space was allocated to this detail and we will remove it in the next version of our paper. Those considerations were a consequence of the initialization bug of the $D$ matrix we had in our code, c.f. the [global answer](https://openreview.net/forum?id=Wa1GGPqjUn&noteId=HB3E5CrCGS) for more details, and do not hold anymore.
>
> > Can the authors clarify the evidence for learning of long-range dependence and offer some additional context in line with the first point above? Do any of the results in Table 1 suggest that the model has learned long-range dependencies?
>
> We will discuss in the next version of the paper the required attention span analysis provided in the original LRA paper [24]. This analysis shows that transformer models need to attend to past inputs on the order of hundreds for the tasks we considered. Our online-learned models greatly outperform transformers, which suggests that long-range dependencies are being captured. While based only on these results we cannot entirely rule out that our models could be making better use of shorter contexts, it seems unlikely that this is the case for LRUs; we will nonetheless add a disclaimer for this point.
>
> > Can the authors share their reasons for limiting evaluation to a subset of the LRA benchmark?
>
> We now ran an additional set of experiments on the IMDB LRA benchmark, where we observe the same overall pattern of results, with our method coming closer to BPTT than to spatial backpropagation in terms of performance (cf. rebuttal results pdf). We didn’t run experiments on the other datasets of the LRA benchmark. The gap between linear RNNs and LRUs is close to 0 in the retrieval dataset (see Table 8 in [24]) so differences between online learning methods are likely to be small. For the pathfinder tasks, we could not get more than chance level with BPTT and the online version of the loss, suggesting that further algorithmic / architecture developments are needed before being able to learn those tasks online. We will add a short comment explaining this rationale to the next version of the paper.
>
> > Typically, “online” learning refers to contexts in which data continuously streams from a source and low-cost updates are made in real time, after which an observation is not revisited. Do the experiments here follow this setup, or are multiple forward passes over the data required? If the latter, what is the advantage in practice of this method over BPTT?
>
> We indeed allow for multiple passes over a finite training set, as is customarily done in the approximate RTRL literature. We opted for this standard setup to make it easier to gauge how close our method can now get to conventional, offline gradient-based learning via BPTT, which is traditionally far better than approximate RTRL. We will clarify this rationale in the next version of the paper.
>
> We are fully available to answer any further questions you may have during the discussion period. We note that we ran additional experiments triggered by other reviews, that we collected in the global response ([link](https://openreview.net/forum?id=Wa1GGPqjUn&noteId=HB3E5CrCGS)).

---

> > ### Comment · Reviewer_XofY · 2023-08-18
> >
> > Thanks to the authors for their engagement and discussion. I have read their replies above and the discussions elsewhere, and together these appear to have led to some fruitful improvements or clarifications. Overall I feel that my original score of 6 is reasonable for this work.

---

### Official Review · Reviewer_Z8SA · 2023-07-07

**Soundness:** 3 good
**Presentation:** 2 fair
**Contribution:** 3 good
**Rating:** 6
**Confidence:** 4

**Summary:**

This paper introduces an online learning algorithm for recurrent neural networks, particularly targeting the learning of long-range dependencies. It builds upon the linear recurrent units and independent recurrent modules in multi-layer networks with complex-valued neural activities. The online update approach, originally proposed in the SnAp-1 algorithm for real values, is here extended to handle complex-valued neural activities the reduces the memory and computational requirements of training. As a result, this approach outperforms both spatial (online) backpropagation and prior approximate real-time recurrent learning approaches in copy task benchmark with sequence of 50 while having marginal improvement over spatial backpropagation for the long-range arena benchmarks.

**Strengths:**

The proposed online learning algorithm effectively optimizes both single-layer and multi-layer recurrent neural network architectures. By transitioning from real-valued to complex-valued neural activities and employing a diagonal recurrent connectivity matrix, it maintains good performance on long-range temporal tasks and enhances online gradient estimation. It accomplishes exact online gradient computation within a single layer using only double the resources required for inference. In managing multi-layer linear recurrent units, proposed approach adeptly approximates backpropagated errors and augments hidden states, mitigating the memory-scaling issue inherent in real-time recurrent learning. Furthermore, it can be thought of a refinement to the SnAp-1 algorithm that can handle independent recurrent units online, offering robust theoretical guarantees despite the approximation used, and accurately computes gradients for all layers, thereby improving gradient alignment and boosting overall performance.

**Weaknesses:**

* The proposed algorithm, while innovative, has some limitations acknowledged by the authors. Its approximation of the error variable, δ, can degrade over time, particularly when the neural activity values converge around 1, leading to a disregard of future error information. This approximation error is compounded when backpropagated through multiple layers, resulting in only partial error signal backpropagation. Hence, the algorithm's effectiveness can be reduced, particularly when managing complex dynamics across many layers.

* The novelty of the algorithm is also somewhat constrained, as it builds upon existing concepts of Linear Recurrent Units (LRUs) and online Recurrent Neural Network (RNN) training methodologies, adapting the mechanisms of the SnAp-1 algorithm to handle complex-valued entities in independent recurrent units.

* The proposed algorithm's performance, while impressive on tasks like the copy task, still lags significantly behind full backpropagation-through-time (BPTT) on longer sequence tasks such as the long-range arena benchmarks with a sequence length of 1000.

* Finally, the methodology employed for hyperparameter selection — using the hyperparameters from BPTT as a basis — isn't practically applicable in an online learning scenario where BPTT won't be available.


**Questions:**

* How would the hyperparameters be optimized for the online learning setting where BPTT results are not available?

* There seems to be a significant drop in accuracy in LRA benchmarks compared to the BPTT. Does this mean that the proposed approach doesn’t scale well to larger sequences? If so, it would be good to discuss the typical sequence lengths that this approach would be ideal for.


**Limitations:**

limitations of the works has been discussed briefly, but not the potential negative societal impact

---

> ### Author Rebuttal · Authors · 2023-08-09
>
> Thank you for the useful questions and comments. We reply below to each of them.
>
> > The novelty of the algorithm is also somewhat constrained, as it builds upon existing concepts of Linear Recurrent Units (LRUs) and online Recurrent Neural Network (RNN) training methodologies, adapting the mechanisms of the SnAp-1 algorithm to handle complex-valued entities in independent recurrent units.
>
> Thank you for raising this point, which made us realize that we were not precise enough in our comparison with SnAp-1. We will update our manuscript with a detailed discussion, cf. global response [link](https://openreview.net/forum?id=Wa1GGPqjUn&noteId=HB3E5CrCGS). Besides being able to learn complex neurons, our rule handles multilayer recurrent neural networks (RNNs) differently, something that we failed to point out previously. More concretely, SnAp-1 does not send detailed spatial credit assignment information in deep RNN models such as the LRU. At best it only broadcasts a global error backwards through skip connections, and in the worst case for networks without skip connections it leads to identically zero updates to deep hidden recurrent neurons (that do not directly influence the loss). Getting a learning signal for such neurons requires raising $k$ in SnAp-$k$, which comes at additional computational and memory costs. In contrast to SnAp, by combining forward sensitivity propagation with spatially backpropagated errors, our hybrid learning rule delivers detailed learning signals even to hidden recurrent neurons, while retaining the cost of a single inference pass.
>
> > The proposed algorithm's performance, while impressive on tasks like the copy task, still lags significantly behind full backpropagation-through-time (BPTT) on longer sequence tasks such as the long-range arena benchmarks with a sequence length of 1000. [...] There seems to be a significant drop in accuracy in LRA benchmarks compared to the BPTT. Does this mean that the proposed approach doesn’t scale well to larger sequences? If so, it would be good to discuss the typical sequence lengths that this approach would be ideal for.
>
> We reran our experiments after fixing a number of bugs (detailed in the global response, [link](https://openreview.net/forum?id=Wa1GGPqjUn&noteId=HB3E5CrCGS)) and retuned hyperparameters for every method separately, no longer perturbing around the hyperparameters that were optimal for BPTT. This widened the gap between our algorithm and spatial backpropagation, in particular on the long-range arena benchmarks, cf. rebuttal results pdf. We note that these results are quite strong in absolute terms, greatly outperforming transformer models, which in turn already require attention spans on the order of hundreds of tokens for these tasks (cf. [24]).
>
> Moreover, to give a better sense of the significance of our results, we performed new sequential CIFAR experiments now using dense stacked linear recurrent networks (the fully-connected counterpart of the diagonal LRU) trained online. These dense models are significantly outperformed by our online-learned diagonal LRU networks.
>
> > Finally, the methodology employed for hyperparameter selection — using the hyperparameters from BPTT as a basis — isn't practically applicable in an online learning scenario where BPTT won't be available. [...] How would the hyperparameters be optimized for the online learning setting where BPTT results are not available?
>
> While we fully agree with the reviewer that ultimately this question will need to be answered, we feel that it is too difficult of a challenge on its own, and out of the scope of the present study. We will add a remark to the experimental section of our paper emphasizing that our results should read as an upper bound obtained using hindsight knowledge of the best hyperparameters, as is customarily done in the approximate RTRL literature. We will further note that complementary techniques developed in the continual and broader online learning literature may be used to help tune hyperparameters in an online, adaptive fashion.
>
> We are pleased to answer any further questions that you may have during the ensuing discussion period. We note that we ran additional experiments triggered by other reviews, that we collected in the global response ([here](https://openreview.net/forum?id=Wa1GGPqjUn&noteId=HB3E5CrCGS)).

---

### Author Rebuttal · Authors · 2023-08-09

We thank the reviewers for the useful comments and questions. The points raised in the reviews led us to run several new experiments and to clarify certain aspects of our manuscript. We believe that these changes have significantly improved our paper. We summarize below the major changes and reply individually to each reviewer in separate threads.

**Detailed discussion of SnAp-1 and e-prop.** Thanks to the reviewers we realized that we did not explain in sufficient detail the relationship between SnAp-1 [15] and e-prop [13] and our learning rule. While both rules are definitely closely related to ours (and we will stress this accordingly), we will now explain the two innovations introduced in our paper: (1) the extension to the complex domain, and (2) our hybrid gradient computation strategy that leverages spatial backpropagation (together with forward sensitivity recursions) to assign credit over multiple recurrent layers. In particular, a discussion of (2) will be added to the next version of the paper. As we explain below, this innovation enables sending detailed spatial credit assignment information over networks with multiple layers.

For a network comprising a single layer of IRMs, SnAp-1, e-prop, our rule, and in fact exact RTRL, all become identical. However, the important multilayer case is handled differently. On the one hand, SnAp-$k$ uses the standard RTRL gradient decomposition $\nabla_\theta L=\sum_t \frac{\partial L_t}{\partial h_t}\frac{dh_t}{d\theta}$, carrying forward in time an approximation of $\frac{\mathrm{d} h_t}{\mathrm{d} \theta}$ that improves with increasing $k$. The partial derivative $\partial L_t / \partial h_t$ is identically zero for neurons that are not directly connected to the output loss; for hidden recurrent neurons $l$ layers away from the output to receive a learning signal, $k$ has to be at least $l+1$. Thus, for networks with deep recurrent neurons, SnAp-1 would only learn the last recurrent layer. Adding skip connections would only ameliorate this problem, as SnAp-1 would only broadcast a global error through that pathway alone. Such differences do not matter for the shallow networks the SnAp paper focuses on, but become important for us, as we move to deep networks. On the other hand, e-prop starts from the same decomposition as us, $\nabla_\theta L = \sum_t \frac{dL_t}{d h_t}\frac{dh_t}{d\theta}$, also noting that $\frac{d L_t}{dh_t}$ cannot be computed causally. The e-prop rule is then derived by approximating $\frac{d L_t}{dh_t}$ by $\frac{\partial L_t}{\partial h_t}$, which then leads to the same situation as in SnAp-1. We, instead, opted for approximating $\frac{d L_t}{dh_t}$ using spatial backpropagation, enabling credit assignment over multiple IRM layers, and therefore sending detailed errors to models with hidden, deep recurrent neurons. We will add this discussion to the next version of our paper.

**Linear RNN results on sequential CIFAR**. We ran a new set of experiments on sCIFAR, replacing LRUs by dense linear RNN layers, cf. pdf. We find that online-learned complex diagonal networks greatly outperform their dense counterparts.

**Addition of GRU baselines.** We further investigated parameter-matched (wide) single-layer GRU networks on the copy task, essentially the architecture used in the SnAp paper [15], as well as multilayer GRU networks, cf. pdf. We ran these experiments to investigate the impact of depth on approximate RTRL in another popular, powerful architecture studied in previous work [9,15]. To train such networks online, we used SnAp-1 for single-layer GRUs, and our hybrid combination of spatial backpropagation with forward sensitivities for multilayer GRUs, cf. point on SnAp-1/e-prop. We find that neither shallow nor multilayer GRUs match the performance of LRU networks trained with our online learning rule. Moreover, only BPTT can take advantage of multiple GRU layers, whereas online GRU learning stagnates at single-layer performance. This again confirms the importance of the IRM design motif in enabling accurate online gradient estimation. We are currently running similar experiments on the sCIFAR benchmark and will add the results to the next version of the paper.

**New IMDB LRA benchmark.** We ran one more benchmark from the LRA suite, IMDB, to confirm that our trend of results remains (cf. pdf).

**Overall improvement in results.** We noticed some bugs in our code and brought some additional improvements after the submission. We here summarize the changes we made and the impact they have on our results:

- We were initializing the $D$ matrix without proper normalization; we now normalize it. We  observed that, thanks to this change, the explosion phenomenon in linear RNNs no longer appears.
- Beforehand, we ignored the normalization factor $\gamma$ in the backward pass. We fixed this.
- Previously, for all experiments in the LRA benchmark, the prediction for time step t was created by taking the softmax of the current encoding. We now change it to be the softmax of the cumulative mean of all previous encodings, and block gradients for all previous time steps to ensure that our learning rule remains causal. This way, we get closer to the non-causal mean-pooling usually used.

These changes led to an overall improvement in all methods, particularly the online & LRU ones, c.f. pdf.

Interestingly, we additionally observed that it is now possible to reach a 0 training loss on the copy task when training a 4-layer deep LRU network with our learning rule for 250 epochs. This goes against the result that uniformly-biased gradient descent doesn't converge to a global optimum (D’Aspremont, SIAM Journal on Optimization, 2008) and may suggest that the bias of our learning rule has some special structure.

We also retuned hyperparameters (learning rate and weight decay) by performing separate grid searches for every method; details will be shared in the next version of the supplement.

---

### Decision · Program_Chairs · 2023-09-21

**Decision:**

Accept (poster)

**Comment:**

This paper introduces an online learning algorithm for recurrent neural networks, particularly targeting the learning of long-range dependencies. In this regards, the authors show that applying an online learning algorithm to independent recurrent modules of linear recurrent units, drastically reduces the algorithm’s computational and memory requirements. All of the reviewers seem to be on the positive side. We thank the authors for providing a detailed response to reviewer concerns. Please include the extended discussion with prior works (i.e. SnAp-1 and e-prop) and all the additional experimental results in the final version of the paper.